# Anti-SARS-CoV-2 Antibody Development over Four Years in Blood Donors

**DOI:** 10.3390/v17101292

**Published:** 2025-09-24

**Authors:** Christoph Niederhauser, Nadja Widmer, Caroline Tinguely, Franziska Suter Riniker, Stefano Fontana, Andreas Buser, Sophie Waldvogel, Jutta Thierbach, Max Züger, Peter Gowland

**Affiliations:** 1Interregional Blood Transfusion SRC, 3008 Bern, Switzerland; nadja.widmer@itransfusion.ch (N.W.); caroline.tinguely@itransfusion.ch (C.T.); stefano.fontana@itransfusion.ch (S.F.); peter.gowland@sunrise.ch (P.G.); 2Institute of Infectious Diseases, University of Bern, 3001 Bern, Switzerland; franziska.suter@unibe.ch; 3Faculty of Biological Medicine, Lausanne University Hospital, 1005 Lausanne, Switzerland; 4Blood Transfusion Service SRC Della Svizzera Italiana, 6900 Lugano, Switzerland; 5Blood Transfusion Service SRC Beider Basel, 4031 Basel, Switzerland; andreas.buser@usb.ch; 6Blood Transfusion Service SRC Genève, 1211 Genève, Switzerland; sophie.waldvogelabramowski@hcuge.ch; 7Blood Transfusion Service SRC Nordostschweiz, 9000 St. Gallen, Switzerland; jutta.thierbach@blutspende-sg.ch; 8Blood Transfusion Service SRC Thurgau, 8500 Frauenfeld, Switzerland; max.züger@blutspende-sg.ch

**Keywords:** SARS-CoV-2 virus 1, blood donors, anti-S, anti-NCP, seroprevalence epidemiology

## Abstract

The blood donor population has great potential to serve as a sentinel system for the general population. The data presented in this study of a survey of the seroprevalence of Anti-SARS-CoV-2 antibodies in Switzerland are an illustrative example of this possibility. From March 2020 to January 2021, the increase in antibody seroprevalence for both Anti-NCP and Anti-S was only very moderate, up to ~17%. In May/June 2021, the Anti-NCP seroprevalence was 21.6% and the S-seroprevalence 59.4%, respectively, indicating only a moderate natural infection rate. The Anti-seroprevalence rate was in good agreement with the first vaccination campaign launched in winter 2020/21. The dramatic increase in the antibodies against the NCP protein (74.6%) since November/December 2021 to June–August 2022 was simultaneous with the appearance and rapid spread of the Omicron SARS-CoV-2 variant in Switzerland and the abolition of compulsory mask wearing in public spaces. At the end of 2022, 99.0% of the blood donor population already developed antibodies against the S protein and 83.9% against the NCP. One year later, after the official termination of the pandemic, these seroprevalences were even higher, 99.8% for the Anti-S and 95.0% for Anti-NCP. This increase for both of them was in accordance with the vaccination campaigns, the abolition of mask wearing, and the spread of the Omicron variant. These data show how the blood donor population can be used to represent the infection surveillance of the general population of a region or country.

## 1. Introduction

In Switzerland, the first coronavirus disease 2019 (COVID-19) case was registered on 25 February 2020 [1] and the first COVID-19 wave occurred in late March and ended by late May 2020 [2,3]. At that time, the Swiss authorities adopted a wide range of lockdown protective measures covering many sectors (i.e., health, economy, mobility, employment) to contain SARS-CoV-2 rapid spread, protect civilians, and mitigate the economic burden of the pandemic to the state [4].

Most individuals with a confirmed SARS-CoV-2 infection developed measurable antibodies, as did those individuals who were vaccinated. Seroprevalence studies can therefore help to identify infections missed by mandatory reporting and support decisions on infection control measures. The majority of seroprevalence studies performed addressed only certain smaller region or a specific part of the population or certain cohorts of patients and were often limited to a single sampling or a few time points, or were partly also very resource intensive [5,6,7,8,9,10].

The purpose of serosurveillance studies is to study the percentage of a population which has antibodies against an infectious disease agent (IDA). Investigating the seroprevalence for SARS-CoV-2 in the general population was of the utmost importance to estimate the circulation of the virus in any geographical area. Such surveys reflect the measures taken at different timepoints and provide a more comprehensive representation of the burden of disease and population-level immunity. Ideally, serial sampling from the general population would provide the most generalizable results, but this approach is both time consuming and expensive.

In the case of the SARS-CoV-2 scenario, a differentiation between a natural infection and vaccination-induced immunity should be known. The blood donor population shows great potential to serve as a sentinel system for the general population. Therefore, surveys of the prevalence of Anti-SARS-CoV-2 antibodies in the blood donor population were conducted in different European countries or regions and provided useful epidemiological information for monitoring the epidemic over time and estimating the level of herd immunity of these within a specific time frame and in relation to an entire country [11,12,13,14,15,16,17,18]. The important weakness of the blood donor population as a sentinel for the general population is the potential for certain bias due to the “healthy donor effect.” Blood donors are generally healthier than the general population, as they must meet specific health criteria to donate. This means that studies using blood donor samples may not accurately reflect the broader population’s health status or disease prevalence.

The basic tool for investigating the seroprevalence of infectious disease agents is the use of highly sensitive and specific assays for the specified infectious disease agent. At the beginning of the SARS-CoV-2 pandemic, probably many of these assays were not really of high quality in terms of their specificity and sensitivity. In general, only a small number of samples were used in these studies to determine these specificities and sensitivities. Prior to testing our blood donors during the pandemic, we thus first compared six different Anti-SARS-CoV-2 assays that were commercially available at that time point. The aim was to use the tests that had the highest sensitivity and the best specificity for the seroprevalence study. The data collected from this comparison lead to the decision for two assays, Anti-SARS-CoV-2 Spike protein (Anti-S) and Anti-SARS-CoV-2 Nucleocapsid protein (Anti-NCP), both from Roche Diagnostics, to be used for the current study.

The present study focused on a nearly four-year course (March 2020 to January 2024) of Anti-SARS-CoV-2 seroprevalence against the spike protein (S) and nucleocapsid protein (NCP) in Swiss blood donors, sampled from the whole country, differentiating between natural infections and vaccination-induced immunity, and in addition, on gaining a better knowledge of our blood donor population and also making a certain statement regarding the general population.

## 2. Materials and Methods

### 2.1. Preliminary Studies of Pre-Pandemic Blood Donor Samples

Comparison of Six Different Assays

The 4044 anonymised, undiluted (frozen at −30 °C and thawed once for the analysis of the six assays) blood donor samples were collected from March to June 2018; a period prior to the first report of the new SARS-CoV-2 infections was used for the comparison of all six SARS-CoV-2 antibody assays. These samples were used to evaluate the assay specificities of the SARS-CoV-2 ELISA tests included in the comparison. The assay sensitivity of the Roche Elecsys SARS-CoV-2 NCP test was taken from the study conducted by Riester and colleagues: 93.61% (95% CI 89.51–96.46) on samples collected at least 14 days post-PCR confirmation of SARS-CoV-2 infection [19]. The following 6 commercially available assays were investigated: Anti-NCP assays: Platelia SARS-2 total antibody on the Evolis (Bio-Rad Laboratories Inc., Marnes-la-Coquette, France), Elecsys Anti-SARS-CoV-2 on the cobas e 801 instrument (Roche Diagnostics, Rotkreuz, Switzerland); SARS-CoV-2 IgG on the Architect (Abbott, ME, USA); Anti-SARS-CoV-2 NCP (Euroimmun, Lübeck, Germany) on the Evolis Analyzer and for Anti-S: Liaison SARS-CoV-2 S1/S2/IgG on the LIAISON XL Analyzer (Diasorin, Saluggia, Italy), Anti-SARS-CoV-2 IgG (Euroimmun, Lübeck, Germany) on the Evolis Analyzer. All the tests were performed according to the package inserts.

### 2.2. Seroprevalence Study over Nearly Four Years

#### 2.2.1. Donor Population

A total of 17,329 random residual plasma samples from voluntary non-renumerated blood donors were analysed. The age of the donors ranged from 18 and 75 years. The samples were collected in seven different regions of Switzerland (Eastern Switzerland; Thurgau, St. Gallen; Northern Switzerland; Basel, Middle Switzerland; Bern, Western Switzerland; Vaud and Geneva; Southern Switzerland; Ticino and Wallis) over eight consecutive time points (March 2020, June–August 2020, January 2021, May–June 2021, November–December 2021, June–August 2022, October–December 2022 and January 2024). Between 1953 and 2319 blood donor samples were investigated at any of these eight time points (Table 1). The EDTA plasma blood donor samples were taken retrospectively from archive plates stored at −20 °C, as well as fresh plasma samples for the samples taken between October–December 2022. All donors were symptom-free at the time of donation.

#### 2.2.2. Serological Analysis

SARS-CoV-2 NCP (Anti-NCP) and SARS-CoV-2 S (Anti-S) antibody testing was performed on all samples with the Elecsys Anti-SARS-CoV-2 electrochemiluminescence immunoassay (ECLIA; total antibody assay against the NCP) and the Elecsys Anti-SARS-CoV-2 S electrochemiluminescence immunoassay (ECLIA; total antibody assay against the S protein) on the cobas e 801 instruments, according to the manufacturer’s instructions (Roche Diagnostics, Rotkreuz, Switzerland).

#### 2.2.3. Measures Ordered by the Federal Office of Public Health (FOPH)

In short, the most important mandatory measures set by the FOPH were the following: mandatory lockdowns from March–June 2020 and December 2020–February 2021, compulsory mask wearing in public spaces, the time points for the 1st and 2nd SARS-CoV-2 vaccination campaigns and later the booster vaccination campaign, and finally the removal of the compulsory mask wearing situation [4,20,21].

From Thursday, 17 February 2022, face masks and COVID-19 certificates were no longer required to enter shops, restaurants, cultural venues, and events and other public settings. The requirement to wear masks in the workplace and the recommendation to work from home also ended. The sole requirement that remained was to isolate in the event of a positive test and to wear masks on public transport and in healthcare institutions in order to protect high-risk individuals. At end of March 2022, all governmental requirements were dropped [20].

#### 2.2.4. Vaccination Strategy in Switzerland

In Switzerland, two different SARS-CoV-2 vaccines were available during the study period (Moderna and Pfizer/BioNTech). On 19 December 2020, the Swiss Agency for Therapeutic Products (Swissmedic) approved the Pfizer/BioNTech vaccine, and on 12 January 2021 the second COVID-19 vaccine from Moderna was approved by Swissmedic [22,23]. As of 14 November, there have been 6,123,678 people who have taken the first dose of coronavirus vaccine in Switzerland [24].

As both these vaccines only target the S protein, Anti-S antibodies alone were considered a response to vaccination, whereas antibodies to both the nucleocapsid protein and the S protein were formed in response to natural infection with or without symptoms. Thus, an Anti-S without Anti-NCP response was considered a proxy for vaccination response, at least when natural infection prevalence is low. As part of the longest running SARS-CoV-2 seroprevalence study in Switzerland, our study monitored vaccine-generated Anti-S and natural infection-generated Anti-NCP development.

#### 2.2.5. Ethics Committee

All the samples were anonymised prior to testing. The whole study was approved by the following cantonal ethics commissions (EC): Bern, Geneva, Northwest Switzerland, Northeast Switzerland, Ticino, and Vaud. The lead ethics commission was from EC Bern (2021-02502).

## 3. Results

### 3.1. Assay Specificity and Sensitivity of the Assays Used for the Current Study

A total of 4044 pre-pandemic blood donor samples were selected from March–June 2018. They were tested with 6 different commercial SARS-CoV-2 assays and subsequently analysed for their test specificities (Table 2 and Figure A1a–f in Appendix A). The Elecsys Anti-SARS-CoV-2 S test from Roche Diagnostics showed a sensitivity in samples taken ≥14 days post-PCR (*n* = 240) of 97.92% (95% CI: 95.21–99.32) [25]. The Elecsys Anti-SARS-CoV-2 was chosen as the NCP assay for the current seroprevalence study of blood donors. Later, the Elecsys Anti-SARS-CoV-2 S electrochemiluminescence immunoassay was chosen for the S antigen analysis. Both assays ran on the Roche Diagnostics cobas e 801 instrument/Roche Diagnostics (Switzerland) Ltd., Rotkreuz, Switzerland.

### 3.2. Seroprevalence Study Results

A total of 17,329 random blood donor samples, from eligible donors fulfilling all criteria as blood donors, over 47 months (March 2020 to January 2024), from seven different Swiss regions were investigated over eight consecutive time points. The lowest Anti-SARS antibody percentages were observed during the first period in March 2020, before the start of the SARS-CoV-2 epidemic in Switzerland: Anti-NCP 0.27% (95% CI, 0.05–0.49%) and Anti-S 0.32% (95% CI, 0.08–0.57%) (Table 1, Figure 1 and Figure 2). At the second time point between June and August 2020, the corresponding values were Anti-NCP 4.31% (95% CI, 3.46–5.16) and Anti-S 4.27% (95% CI, 3.42–5.11%), and at the third time point in January 2021, both antibodies’ levels had risen to 16.39% (95% CI, 14.83–17.96%) for Anti-NCP and 16.80% (95% CI, 15.23–18.40%) for Anti-S, respectively. Five months later, between May and June 2021, the outcome of the first SARS-CoV-2 vaccine campaign was obvious. The Anti-S with 55.71% (95% CI, 53.69–57.74%) was significantly higher than the Anti-NCP, which showed only a slight increase to 21.09% (95% CI, 19.43–22.75%). Data from the fifth time point showed an even better success of the vaccine campaign. The Anti-S had increased to 90.54% (95% CI, 89.30–91.78%), whereas the Anti-NCP did not significantly increase from the previous time point: 22.0% (95% CI, 20.25–23.76%). At the sixth time point taken between June–July 2022, the highly infectious SARS-CoV-2 Omicron variants had become established in Switzerland. In January 2022, these variants were identified in 95% of the viruses isolated (Figure 2). This situation was portrayed in the dramatic increase in Anti-NCP of 74.4% (95% CI, 72.53–76.19%), as well as in the Anti-S value of 98.22% (95% CI, 97.67–98.78%) (Figure 2). At the next time point from October to December 2022, the Anti-S remained high with 99.04% (95% CI, 98.64–99.45%) and the Anti-NCP seroprevalence raised once again to 83.88% (95% CI, 82.34–85.42%). The last time point at which samples were collected was in January 2024, after the pandemic seemed to have run its course. Nevertheless, a further increase for both the Anti-S at 99.77% (95% CI, 99.57–97.97%) and astonishingly also the Anti-NCP, at 94.98% (95% CI, 94.97–94.99%), was observed. This indicates that the virus was still circulating in the population and infecting naïve, vaccinated and previously infected individuals.

The average Anti-NCP MOC (multiple over cutoff) values over the eight time points were in the range of 23.04, from the first time point to 74.30 at the second time point (Figure 3). Later these values fluctuated between 33.53 and 58.68. Surprisingly, the highest value was seen at the last time point of January 2024 with 121.28 (Figure 3). CI, confidence interval.

## 4. Discussion

**General:** The serological surveillance of antibodies has been shown to be an effective method of monitoring the spread of infections in a population and has provided vital information on the circulation of diseases in a population, including information on those individuals who may have experienced mild or asymptomatic disease [26,27,28,29,30]. In general, the blood donor population can be used as a proxy for the healthy adult population and therefore can complement surveillance data from notification systems of confirmed cases. Blood donor testing has proven to be a valuable tool in the study of asymptomatic or subclinical conditions and can be used for population-based incidence and prevalence monitoring in a relatively cost-effective manner compared with longitudinal cohort studies, which are conducted normally on much smaller sample sizes and are exponentially more expensive and difficult to perform. As blood donor cohorts are easily and routinely available, serial studies allow for the assessment of seroprevalence over time, including during and between infection waves. Numerous SARS-CoV-2 seroprevalence studies in blood donor cohorts have contributed significantly to the global and regional understanding of the SARS-CoV-2 pandemic, providing information on breakthrough and reinfection rates [14,31,32,33]. Many blood services around the world capitalised on their infrastructure to quickly start SARS-CoV-2 seroprevalence studies to inform future public health policies. By June 2020, a short three-month period after the pandemic was declared, 32 of 48 countries surveyed had initiated SARS-CoV-2 seroprevalence studies by blood transfusion services [34]. The strengths of blood donors as a substitute for the general population are the following: (i) about 90% of donors in Switzerland donate repeatedly; therefore, these donors can form a cohort for on-going monitoring, (ii) data including demographic variables (e.g., age, sex, postal code, and ethnicity), current medications, recent vaccinations, and recent travel history are collected via the routine donor history questionnaire and could be constantly available, (iii) the near-national reach of blood services’ daily collections and laboratory capacity can be used to rapidly survey pathogens at a relatively low cost, (iv) blood donor services also undertake unique lookback processes (investigating recipients of a test-/disease-positive donor) and traceback processes (investigating donations and donors from a disease-/test-positive recipient) for blood recipients or donors with a suspected blood-borne infection. These processes could be adapted to support further active surveillance and (v) for national surveillance activities, and there are substantial advantages over other sources of healthy individuals, for example, patient testing and pregnancy screening programmes, which are generally local rather than national.

**Selection of suitable tests:** At the beginning of the pandemic, it was very important that highly specific but also sensitive SARS-CoV-2 assays were characterised to generate valuable data. For this reason, we compared six SARS-CoV-2 antibody tests that were commercially available at that time point. Based on these data and the already published data in terms of the specificity and sensitivity of SARS-CoV-2 antibody assays, we then decided in favour of the assays from Roche Diagnostics [19,25,35]. The Roche assays, both the Anti-S and the Anti-NCP, were among the ones with the highest performance in terms of specificity and sensitivity (Figure A1a–f, Addendum). At the beginning of the pandemic, unfortunately, the published specificity studies had only been conducted on a small number of pre-pandemic samples and therefore often unreliable specificity values were known. Later on, studies with a greater number of samples were performed. Lewin and colleagues highlighted the big diversity of the Anti-SARS-CoV-2 assays used, with a wide range of sensitivities and specificities [34].

**Serology surveillance of blood donors:** Between 1953 and 2319 donations from voluntary blood donors were tested at eight different timepoints (March 2020, June–August 2020, January 2021, May–June 2021, November–December 2021, June–August 2022, October–December 2022 and January 2024). In the initial phase of the pandemic during March 2020, the seroprevalence against both the NCP and S protein in blood donors investigated in the present study was low, up to a maximum of 0.3%. Later, between June–August 2020, an increase to 4.3% in parallel for both parameters (NCP and S) was observed. These low seroprevalences were comparable to other middle-European regions. Several European countries revealed the following seroprevalences: Germany 0.91%; Spain and Italy 0.27–2.7% [13,14,17]. In Switzerland, the cumulative prevalence from different regional investigations rose from 2.3% in June 2020 to 12.2% in mid-December 2020, which was roughly congruent with the data from the blood donor population [36]. There was a modest increase over the summer months, followed by a rapid rise in late 2020. It was estimated that 10.3–14.6% had undergone an infection with SARS-CoV-2 by mid-December 2020 [36]. In our blood donor population, by January 2021, after the second COVID-19 wave (between October and January 2020), an increase in the seroprevalence for both antibodies was also observed (16.4% Anti-NCP and 16.8% Anti-S; Table 1, Figure 1 and Figure 2). This increase can be explained by the relaxation of certain protection measures during early summertime, which in turn allowed more public transport use; therefore, it was often impossible to keep the recommended social distance. In view of this increase in travel and the rising number of new infections from June 2021, the federal council began stepping up protective measures and declared the wearing of masks compulsory on all public transport.

On 19 December 2020, Swissmedic approved the first COVID-19 vaccine for the Swiss population [22]. It was recommended during the first weeks of the vaccination campaign that, initially, vulnerable individuals, those aged over 65 and people who were particularly exposed, such as healthcare workers, should be prioritised for vaccination. Later, the general population below 65 years progressively had access to vaccination at the numerous designated centres. In May–June 2021, approximately six months after the vaccination campaign began, a striking difference was observed between the NCP and S seroprevalences. The S seroprevalence had dramatically increased to 55.9% and the NCP seroprevalence only moderately increased (21.1%; Table 2 and Figure 1 and Figure 2), which clearly reflected the success of the vaccination campaigns in increasing immunity in the population, thus hopefully protecting them from severe infections. Blood donor samples taken during November/December 2021 revealed that Anti-NCP seroprevalence was practically at the same level as at six months previously (22.0%). This further reinforced the evidence that the measures taken by the federal authorities to restrict SARS-CoV-2 infections and the massive vaccination campaigns were successful. At this time point, 90.5% of the blood donors had already had antibodies against the S protein.

The emergence of the first SARS-CoV-2 Omicron variant (B.1.1.529) during November 2021 dramatically changed the epidemiology of COVID-19, with a rapid upsurge of cases globally [37] (Figure 2). This highly contagious variant was first detected in Switzerland on 18 November 2021 and spread rapidly. On 1 December 2021, the federal office of public health reported that 4.6% of all isolates typed were this Omicron variant. By mid-December, this had increased to 26%; then, at the beginning of January 2022, more than three quarters were found to be this variant (77%), and finally 2 weeks later >95% typed viruses were shown to be the Omicron B.1.1.529 variant [38].

Due to the success of the vaccination campaign, the federal authorities started relaxing the protection measures on 1st February 2022. This included opening shops, restaurants, cultural venues, and publicly accessible facilities and events. They were open to people without a mask but in possession of a vaccination certificate. The requirement to wear masks at work and the recommendation to work from home were also lifted. The obligation to wear masks on public transport and in healthcare facilities remained in place for some time, as well as an isolation requirement for those individuals who tested SARS-CoV-2 antigen- or PCR positive. These measures remained in place until the end of March 2022 to protect particularly vulnerable people; however, after that date, all measures were lifted and the situation returned to normal. But this relaxation of protective measures ignored the fact the Omicron variants were still circulating in the population. This was shown by the significant increase in the Anti-NCP and Anti-S antibodies in our blood donor samples in June–August 2022 (Anti-NCP 74.4%, Anti-S 98.5%; Table 1 and Figure 1). A similar picture was shown in another Swiss cohort study during midsummer 2022 [39]. In this study, the seroprevalence in the general population of 3 of the 27 cantons situated to the south, west and middle of the country was very similar (98.3% in Ticino (95% CI 96.9–99.3%), 98.4% in Vaud (95% CI 97.3–99.3%), and 98.9% in Zurich (95% CI 98–99.5%). These investigations were performed with SARS-CoV-2-specific antibodies against the NCP and S proteins using the sensitive Anti-SARS-CoV-2 Spike Trimer Immunoglobulin Serological (SenASTrIS) test, a Luminex binding assay [40]. Their data represent the combined seroprevalence of both antibodies, the S protein and NC protein, but were congruent with the data from our blood donor population. Another cross-sectional analysis of a prospective, population-based cohort study conducted in Switzerland included 1894 randomly selected 16-to-99-year-old participants from two cantons in March 2022. Of these, 97.6% (95% CI: 96.8–98.2%) had Anti-S IgG antibodies [7]. Thus, by June/July 2022, it appeared that almost the entire population had developed antibodies against SARS-CoV-2, irrespective of age and residency in Switzerland. The seroprevalence of blood donors collected between October to December 2022, however, showed a further increase in the Anti-NCP antibodies (from 74.4% to 83.9%), whereas the Anti-S antibodies remained high (99% versus 98.2%). This is consistent with SARS-CoV-2 contact and even infection, despite vaccination in previously naïve blood donors. As a final sample cohort, we decided to analyse another series of blood donor samples a year later, collected in January 2024, eighteen months after the removal of all the federal restrictions. We hoped to see whether the levels of Anti-NCP and Anti-S antibodies had stabilised, increased, or if there were signs of antibody waning. It was surprising to see that the seroprevalence of Anti-NCP antibodies had further increased from 83.9% to 95.5% (Table 1 and Figure 1). The Anti-S seroprevalence was also slightly higher than a year earlier (99.0% compared to 99.8%).

When analysing and comparing the average MOC values for NCP antibodies for January 2024 in relation to earlier dates, it was striking that the value in January 2024 (121.28 MOC) was the highest of all time points (Figure 3). The mean value was twice that measured in December 2022. Perhaps this indicates that despite a probable waning of NCP antibodies, the frequency of new or re-infections with SARS-CoV-2 was likely to be much higher than assumed. Several studies have shown a substantial decline in Anti-NCP titres, particularly among those individuals with reported mild disease [41,42,43,44,45]. On the other hand, the number of non-reported infections is likely to be high because up to 35.1% of SARS-CoV-2 infections are expected to be asymptomatic or pauci-symptomatic [46]. It is for this reason that serological testing is valuable in confirming previous SARS-CoV-2 infections and has a tremendous importance for the broad surveillance of COVID-19 [47].

This study has its limitations, as blood donors only represent a subset of the healthy adult population. It can be assumed that the donors are more likely to adhere to non-pharmaceutical interventions during the pandemic than the general population due to the “healthy donor effect”, as has been reported in Germany [48]. However, this effect is likely to be less important in monitoring the spread of highly contagious pathogens, such as SARS-CoV-2, where there is a low inherent population immunity than for non-infectious diseases or infectious diseases with low transmissibility. Nevertheless, certain groups of the adult population will be underrepresented in this cohort of samples, for instance, people in care or migrants who are not eligible to donate blood according to the current Haemotherapy Guidelines in Switzerland [49]. On the other hand, our study also has strengths, as follows: i) the frequent and repetitive sampling in many regions of Switzerland for nearly 4 years, ii) the use of a standardised protocol and antibody tests, across sites and time since the beginning of the pandemic until more than year after the official termination of the pandemic. In addition, blood donor specimens are readily available even during lockdowns and can and should be used to support surveillance. This is especially important when emerging infections arise, and there is thus an urgent need to have specific data for that agent. Blood donors are an ideal population for infectious disease surveillance under defined circumstances, and blood transfusion services can partner with public health authorities for informed decision making [50].

Recently, key indicators have been identified to reasonably enable these partnerships [51]. Both public health institutions and blood services should identify ways to cooperate and identify areas in which infectious disease surveillance can be supported by blood donor samples, not only during pandemics. More than 250 seroprevalence studies among blood donors have been identified in 40 countries or territories, with a total of nearly ten million blood donors with CLIA- or ELISA-analysed specimens [52,53]. Seroprevalence estimates ranging from 0.1% in New Zealand in December 2020 to 100% in Scotland in May 2022 have been observed [54].

Figure 2 provides an overview of Anti-CoV-2 NCP and S seroprevalence at the eight time points of the current study. At the same time, the corresponding measures, such as lockdowns, mask wearing, home office obligation, etc., and vaccination campaigns are also described. The start of the spread of the Omicron variant has also been included. This provides a clear overview of the seroprevalence of the S antigen and the NCP antigen at a glance, together with all the measures taken, and presents them in a clear context.

## 5. Conclusions

The COVID-19 pandemic has underlined the importance of serosurveillance as an evidence-based tool to understand population immunity, track viral transmission, and guide public health decision making [54]. During the SARS-CoV-2 pandemic, blood transfusion services had the opportunity to use existing laboratory equipment and reagents to test a large number of individuals, thus collecting valuable data for the management of this infectious disease. Blood donor samples are ideal for these seroprevalence studies, as many samples from various geographical regions are needed rapidly. Data from these studies have shown that such samples are also suitable as sentinels, providing essential information for the entire population in a quick and efficient manner. The blood donor population, however, spans 18 to 75 years old and is not an exact mirror for the general population, especially if you consider the timetable of the COVID-19 vaccination campaign, where the most vulnerable and oldest segment of the population were vaccinated first. The present study focused on the Anti-SARS-CoV-2 seroprevalence against the S and NCP proteins in Swiss blood donors over an extensive 4-year period, from the beginning of the pandemic on March 2020 to 2 years after the pandemic had ended. Monitoring antibodies for NCP and S proteins distinguished between natural infections and vaccine-induced immunity. The increase in antibodies to the S protein in our blood donors was in line with the increase in the vaccinated population. This suggests that in the late phase of the epidemic, the risk of donor-to-donor cross-infection was low. However, the striking increase in antibodies against the NCP protein from spring 2022 can be explained by the successive abolition of mandatory mask use, as well as other measures taken across the entire population in conjunction with the emergence of the highly contagious Omicron variant in the population.

Laboratory monitoring of the SARS-CoV-2 during the pandemic was essential to help implement successful public health measures. The serosurveys identified ongoing infections in both asymptomatic and symptomatic patients. In addition, they complement surveillance data from notification systems of confirmed cases. Blood donor population studies, as reported here, thus serve as a sentinel for public health institutions and will be essential in future ongoing and emerging infections. Blood transfusion services may also be requested to take on new activities such as developing therapeutics or supporting public health surveillance measures. Through activities such as scenario development, tabletop exercises, and drills, blood service providers can prepare for the uncertainties of the next pandemic.

## Figures and Tables

**Figure 1 viruses-17-01292-f001:**
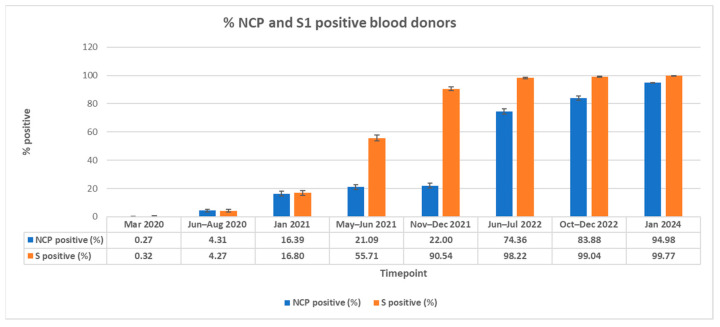
Course of seroprevalences in total over the seven regions during the eight time points for both the Anti-NCP and Anti-S antibody percentages.

**Figure 2 viruses-17-01292-f002:**
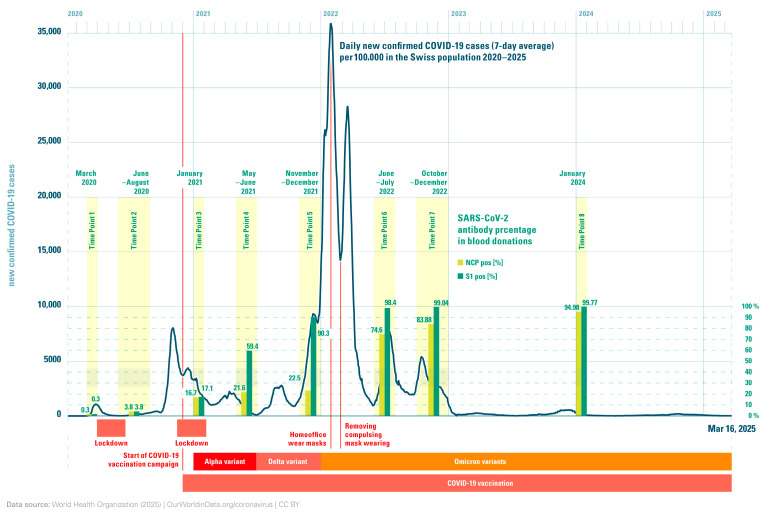
Course of the Anti-SARS-CoV-2 antibodies against the proteins S and NCP in Swiss blood donors in comparison to SARS-CoV-2 variants circulating in Switzerland and the mandatory measures set by the health authorities over 47 months.

**Figure 3 viruses-17-01292-f003:**
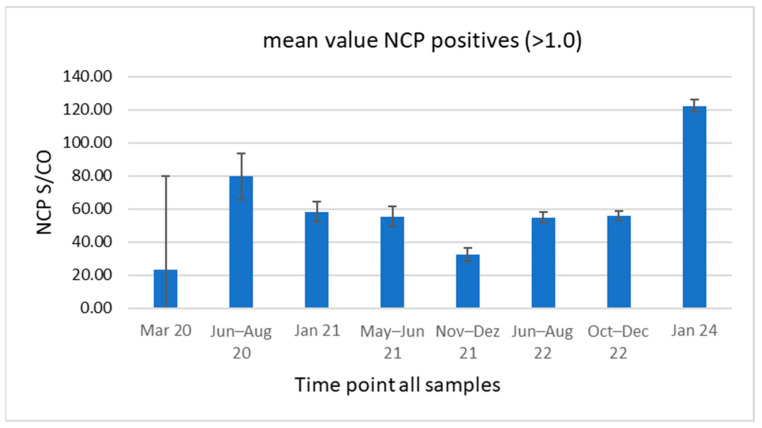
Course of average Anti-NCP MOC values over the total of the seven regions during the eight time points.

**Table 1 viruses-17-01292-t001:** Time course of seroprevalences over the seven regions during the eight time points for both the Anti-NCP and Anti-S antibody percentages.

Timepoints	Donor Samples Tested	NCP Antigen % Seroprevalence (95% CI)	S Antigen % Seroprevalence (95% CI)
March 2020	1953	0.27 (0.05–0.49)	0.32 (0.08–0.57)
June–August 2020	2202	4.31 (3.46–5.16)	4.27 (3.42–5.11)
January 2021	2147	16.39 (14.83–17.96)	16.80 (15.23–18.40)
May–June 2021	2319	21.09 (19.43–22.75)	55.71 (53.69–57.74)
November–December 2021	2145	22.00 (20.25–23.76)	90.54 (89.30–91.78)
June–August 2022	2196	74.36 (72.53–76.19)	98.22 (97.67–98.78)
October–December 2022	2196	83.88 (82.34–85.42)	99.04 (98.64–99.45)
January 2024	2171	94.98 (94.97–94.99)	99.77 (99.57–99.97)

**Table 2 viruses-17-01292-t002:** The 4044 anonymised pre-pandemic donor samples were compared with 6 commercially available SARS-CoV-2 antibody assays.

Assay	Total Number of Samples	Negative Plasma Samples	Positive Plasma Samples	Platform	Antibody/Antigen	Test Principle	Specificity (%) (from PACKAGE Insert)	Specificity (%) (CI) Blood Donors
Biorad: Platelia SARS-CoV-2 Total Ab	4044	4018	26	Euroimmun Analyzer/Evolis	Total Ab/NCP	ELISA	99.51	99.36 (98.95–99.64)
Roche Diagnostics: Elecsys Anti-SARS-CoV-2	4044	4037	7	cobas e 801	Total Ab/NCP	ECLIA	99.81	99.83 (99.57–99.95)
Abbott: Architect SARS-CoV-2 IgG	4044	4022	22	Architect	IgG/NCP	CMIA	99.60	99.46 (99.07–99.71)
Diasorin: LIAISON^®^ SARS-CoV-2 S1/S2 IgG	4044	3991	53	LIAISON XL Analyzer	IgG/S1/S2	CLIA	98.50 *	98.69 (98.14–99.11)
Euroimmun Anti-SARS-CoV-2-ELISA-IgG	4044	4007	37	Euroimmun Analyzer/Evolis	IgG/S1	ELISA	na	99.09 (98.61–99.43)
Euroimmun Anti-SARS-CoV-2-ELISA-NCP IgG	4044	4002	42	Euroimmun Analyzer/Evolis	IgG/NCP truncated	ELISA	99.40	98.96 (98.46–99.33)

The symbols * represent statistically significant differences (*p* < 0.05).

## Data Availability

Available upon request.

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
