# Peer review of "Anti-SARS-CoV-2 Antibody Development over Four Years in Blood Donors"

_viruses, 2025, doi:10.3390/v17101292_

Round 1
Reviewer 1 Report
Comments and Suggestions for Authors
Anti-SARS-CoV-2 Antibody Development Over Four Years in Blood Donors
- Line 58ff: Please be more specific - what exactly are the limitations or constraints?
- Line 74ff: What exactly is the content of these surveys? Where are their weaknesses?
- Line 79-81: Why? Reference is missing.
- Line 81: Have these preliminary studies been published? If yes, the source is missing. If not, they should be understood as part of the analysis and moved to the Results section. Since they reappear in the Methods, they should be framed in the Introduction as part of the analysis and formulated as a hypothesis.
- The study objective is very vaguely formulated. What was the exact aim, in which population? What hypotheses were generated?
- Methods section: The Methods section is disjointed! It needs to clearly explain…
- It remains unclear in the Methods how exactly the tests were compared. What were the outcome measures? Were all samples tested with all assays? Were the samples thawed? Were they diluted?
- Methods: Was the number of samples per time period the same? Were these simply all blood donors? Were there inclusion/exclusion criteria? Was it a random selection?
- As contextual factors for the longitudinal seroprevalence, vaccination rates should definitely be included. If no data on donors’ vaccination status are available, then at least national vaccination rates should be incorporated. The vaccination strategy thus belongs in the Methods, not the Results! Please also add the source of the vaccination strategy, e.g., legal basis.
- “The assay sensitivity of the Roche Elecsys SARS-CoV-2 NCP test was taken from the study conducted by Riester and colleagues: 93.61% (95% CI 89.51–96.46) on samples collected at least 14 days post-PCR confirmation of SARS-CoV-2 infection (19).” → This belongs in the Methods or Discussion. Why was this approach chosen?
- Line 248ff: Please specify the sentence “Numerous SARS-CoV-2 seroprevalence studies in blood donor cohorts have contributed significantly to the global and regional understanding of the SARS-CoV-2 pandemic, providing information on breakthrough and reinfection rates (13, 26-28).” What exactly was their contribution?
- “Blood donors between 18 and 75 years were investigated over 47 months, from the beginning of the COVID-19 pandemic in March 2020 up to January 2024, more than a year after the last protective measures were relaxed.” → This sentence is just repetition and can be removed. In the Results, the findings should be discussed, not repeated or introducing new aspects.
- These low seroprevalences were comparable to other Central European countries such as Germany (0.91%), Spain, and Italy (0.27–2.7%) (12, 13, 16). → Because they were affected by the same pandemic.
- “The first person in Switzerland, a 91-year-old woman” → irrelevant.
- Line 290 ff.: Please add references.
- Line 307 ff.: Please add references.
- The entire Discussion is very imprecise, providing more background data instead of setting the collected results into context and interpreting them concretely.
Minor comments:
- Line 38: June -August → extra/missing space.
- Line 54 to 57: Reference missing.
- Line 74 ff.: Wrong font.
- Line 148: “Table1” → missing space.
- What is the reference for specificity?
- Line 166: Please introduce abbreviations once and then use them consistently. Also, it makes little sense to abbreviate Anti-SARS-CoV-2-Spike IgG as S and Anti-SARS-CoV-2-Nucleocapsid IgG as NCP. Prefer a uniform system, e.g., Anti-S and Anti-N or NCP and SP.
- Table 1: Missing decimal separators.
- Figure 1: Limit axis to 100%; 120% makes no sense. The figure is not easily readable; the confidence intervals should be shown in both directions. Consider connecting the trend with points and CIs instead of bars.
- Figure 2: No data points in the figure; x-axis labels are hard to read due to month overlap.
- Figure 3: The idea is good, but the figure is poorly legible. Very poor resolution; font partly too small and too thin.
Author Response
Manuscript ID: viruses-3768852
“Anti-SARS-CoV-2 Antibody Development Over Four Years in Blood Donors”
Reviewer I
General
- Discussion: Authors should discuss the results and how they can be interpreted in perspective of previous studies and of the working hypotheses. The findings and their implications should be discussed in the broadest context possible and limitations of the work highlighted. Future research directions may also be mentioned. This section may be combined with Results.
We have placed the discussion in a broader context with a paragraph on strengths, weaknesses of the study. In the section conclusion, we have briefly outlined future possibilities for research directions. In addition, the anti-NCP MOC values were also discussed.
Anti-SARS-CoV-2 Antibody Development Over Four Years in Blood Donors
- Line 58ff: Please be more specific - what exactly are the limitations or constraints? Text 58ff has been written more clearly in order to highlight the limitations and constraints. References 4-9 described, for example, studies in only a small region of Switzerland or only specific study populations such as police officers or employees of certain non-health sectors.
- Line 74ff: What exactly is the content of these surveys? Where are their weaknesses?
As requested more explanation for the content of the surveys has been added. In addition, the possible weakness of the blood donor population as a sentinel for the general population is there is a potential for bias due to the "healthy donor effect" . This has been mentioned in the text too.
- Line 79-81: Why? Reference is missing. The number of samples included in many of the earlier published studies were too small to obtain a significant picture. Some References have been added accordingly
- Line 81: Have these preliminary studies been published? If yes, the source is missing. If not, they should be understood as part of the analysis and moved to the Results section. Since they reappear in the Methods, they should be framed in the Introduction as part of the analysis and formulated as a hypothesis. They appear in the section methods, as well in the section results and discussion. A sentence was added as proposed by the reviewer in the section introduction line 92-94,….”The aim was to use the tests for the seroprevalence study that had the highest sensitivity and the best specificity”
- The study objective is very vaguely formulated. What was the exact aim, in which population? What hypotheses were generated? The study focused on blood donors all over Switzerland, over a period of nearly four years, against the S and the NCP proteins in order to differentiate between vaccinated induced and natural infection. We added the following sentence in order to clarify: “and in addition to have a better knowledge on our blood donor population and also a certain statement regarding the general population.”
- Methods section: The Methods section is disjointed! It needs to clearly explain. We have added two subtitles in order to clarify.
It remains unclear in the Methods how exactly the tests were compared. What were the outcome measures? Were all samples tested with all assays? Were the samples thawed? Were they diluted? The sentence has been rephrased in order to clarify. The 4044 undiluted, frozen (at -30°C) EDTA blood donor samples were tested with all 6 commercially available SARS CoV-2 assays.
- Methods: Was the number of samples per time period the same? Were these simply all blood donors? Were there inclusion/exclusion criteria? Eligible for blood transfusion Was it a random selection? Yes random selection, we have added this additional information to the manuscript
- As contextual factors for the longitudinal seroprevalence, vaccination rates should definitely be included. If no data on donors’ vaccination status are available, then at least national vaccination rates should be incorporated. The vaccination strategy thus belongs in the Methods, not the Results! Please also add the source of the vaccination strategy, e.g., legal basis. The section vaccination strategy has been moved from Results to Methods as the Reviewer suggested. In addition a sentence concerning the vaccination strategy has been added.
- “The assay sensitivity of the Roche Elecsys SARS-CoV-2 NCP test was taken from the study conducted by Riester and colleagues: 93.61% (95% CI 89.51–96.46) on samples collected at least 14 days post-PCR confirmation of SARS-CoV-2 infection (19).” → This belongs in the Methods or Discussion. Why was this approach chosen? At that time, we did not have access to adequate patient sample material with the corresponding clinical information. We therefore decided to take the sensitivity data from the literature.
- Line 248ff: Please specify the sentence “Numerous SARS-CoV-2 seroprevalence studies in blood donor cohorts have contributed significantly to the global and regional understanding of the SARS-CoV-2 pandemic, providing information on breakthrough and reinfection rates (13, 26-28).” What exactly was their contribution? Many blood services around the world capitalized on their infrastructure to quickly start SARS-CoV-2 seroprevalence studies to inform future public health policies. By June 2020, a short three month period, after the pandemic was declared, 32 of 48 countries surveyed had had SARS CoV-2 seroprevalence studies initiated by blood transfusion services (O’Brien SF et al. Vox Sang 2021; Lewin A et al. Vox Sang 2021). This sentence has been added.
- “Blood donors between 18 and 75 years were investigated over 47 months, from the beginning of the COVID-19 pandemic in March 2020 up to January 2024, more than a year after the last protective measures were relaxed.” → This sentence is just repetition and can be removed. Sentence has been removed. In the Results, the findings should be discussed, not repeated or introducing new aspects.
- These low seroprevalences were comparable to other Central European countries such as Germany (0.91%), Spain, and Italy (0.27–2.7%) (12, 13, 16). → Because they were affected by the same pandemic. A sentence in order to clarify has been added.
- “The first person in Switzerland, a 91-year-old woman” → irrelevant. Sentence has been deleted.
- Line 290 ff.: Please add references. Ref has been added.
- Line 307 ff.: Please add references. Ref has been added.
- The entire Discussion is very imprecise, providing more background data instead of setting the collected results into context and interpreting them concretely. We have placed the discussion in a broader context with a paragraph on strengths, weaknesses of the study. In the section conclusion, we have briefly outlined future possibilities for research directions. In addition, the anti-NCP MOC values were also discussed.
Minor comments:
- Line 38: June -August → extra/missing space. Thank you for carefully reading. Missing space has been introduced
- Line 54 to 57: Reference missing. Missing reference has been inserted. Important decision from the Swiss Federal Council of Switzerland: https://www.uvek.admin.ch/uvek/de/home/uvek/coronavirus/wichtige-entscheide.html
- Line 74 ff.: Wrong font. Thank you for carefully reading, font has been changed
- Line 148: “Table1” → missing space. Thank you for carefully reading. Missing space has been introduced
- What is the reference for specificity? Specificity data are included in the manuscript, therefore no reference.
- Line 166: Please introduce abbreviations once and then use them consistently. Also, it makes little sense to abbreviate Anti-SARS-CoV-2-Spike IgG as S and Anti-SARS-CoV-2-Nucleocapsid IgG as NCP. Prefer a uniform system, e.g., Anti-S and Anti-N or NCP and SP. We tried to abbreviate and harmonize
- Table 1: Missing decimal separators. The missing decimal separators has been added.
- Figure 1: Limit axis to 100%; 120% makes no sense. The figure is not easily readable; the confidence intervals should be shown in both directions. Consider connecting the trend with points and CIs instead of bars. We changed the figure as far as we could. If we set the axis from 120% to 100%, the image becomes very poor because everything is very cramped. Since we always took measurements at specific points in time, it seems more appropriate to display the data in columns rather than connecting the points. There are usually up to six months between the individual time points of measurement.
- Figure 2: No data points in the figure; x-axis labels are hard to read due to month overlap. We removed the data points in the figure as requested. The monthly overlap have been changed in order to make it more readable.
- Figure 3: The idea is good, but the figure is poorly legible. Very poor resolution; font partly too small and too thin. Thank you for your feedback. We have spoken to the responsible graphic designer, who adjusted then the image to make it easier to read. We believe that it should be clearly legible after digital publication.

Reviewer 2 Report
Comments and Suggestions for Authors
The manuscript reports the results of an epidemiological survey on anti-SARS-CoV-2 antibodies carried out on blood donors samples collected and stored during a period from 2020 to 2024.
The study design is well organized to follow the evolution of the anti-SARS_CoV-2 frequency in the population of blood donors in Switzerland.
The methods are well described with a pre-testing in order to choose the better assays to identify the two differen antibodies produced by natural infections and vaccination.
The data are very useful to follow the immunity evolution in a population exposed to an emerging and unknown infection.
Only few and minor observations:
- In line 84 the studied antibodies are indicated only with acronyms and in line 87 are reported in extenso: it would be better to report the definitions in extenso at first citation in line 84.
- In line 100, after the word “investigated” better a full stop instead of a semicolon.
- The beginning of the vaccination campaign is reported in Fig 3 an in the Discussion chapter. It woul be more appropriate to report this information also in the paragraph 1.2. Vaccination strategy in Switzerland.
Author Response
Manuscript ID: viruses-3768852
“Anti-SARS-CoV-2 Antibody Development Over Four Years in Blood Donors”
Reviewer 2
The manuscript reports the results of an epidemiological survey on anti-SARS-CoV-2 antibodies carried out on blood donors samples collected and stored during a period from 2020 to 2024.
The study design is well organized to follow the evolution of the anti-SARS_CoV-2 frequency in the population of blood donors in Switzerland.
The methods are well described with a pre-testing in order to choose the better assays to identify the two differen antibodies produced by natural infections and vaccination.
The data are very useful to follow the immunity evolution in a population exposed to an emerging and unknown infection.
Only few and minor observations:
- In line 84 the studied antibodies are indicated only with acronyms and in line 87 are reported in extenso: it would be better to report the definitions in extenso at first citation in line 84. Has been changed according to the suggestions.
- In line 100, after the word “investigated” better a full stop instead of a semicolon. Has been changed according to the suggestions.
- The beginning of the vaccination campaign is reported in Fig 3 an in the Discussion chapter. It would be more appropriate to report this information also in the paragraph 1.2. Vaccination strategy in Switzerland. The paragraph ‘Vaccination strategy in Switzerland’ has been moved from the “Results” section to the ‘Materials and methods’ section (as suggested by a reviewer). The approval by the responsible health authority and the start of the vaccination campaign have been added to that paragraph.
